# Mycotoxin Occurrence, Exposure and Health Implications in Infants and Young Children in Sub-Saharan Africa: A Review

**DOI:** 10.3390/foods9111585

**Published:** 2020-11-01

**Authors:** Cynthia Adaku Chilaka, Angela Mally

**Affiliations:** Institute of Pharmacology and Toxicology, Julius Maximilian University of Würzburg, Versbacher Straβe 9, 97078 Würzburg, Germany; mally@toxi.uni-wuerzburg.de

**Keywords:** mycotoxin, occurrence, exposure, child health, sub-Saharan Africa

## Abstract

Infants and young children (IYC) remain the most vulnerable population group to environmental hazards worldwide, especially in economically developing regions such as sub-Saharan Africa (SSA). As a result, several governmental and non-governmental institutions including health, environmental and food safety networks and researchers have been proactive toward protecting this group. Mycotoxins, toxic secondary fungal metabolites, contribute largely to the health risks of this young population. In SSA, the scenario is worsened by socioeconomic status, poor agricultural and storage practices, and low level of awareness, as well as the non-establishment and lack of enforcement of regulatory limits in the region. Studies have revealed mycotoxin occurrence in breast milk and other weaning foods. Of concern is the early exposure of infants to mycotoxins through transplacental transfer and breast milk as a consequence of maternal exposure, which may result in adverse health effects. The current paper presents an overview of mycotoxin occurrence in foods intended for IYC in SSA. It discusses the imperative evidence of mycotoxin exposure of this population group in SSA, taking into account consumption data and the occurrence of mycotoxins in food, as well as biomonitoring approaches. Additionally, it discusses the health implications associated with IYC exposure to mycotoxins in SSA.

## 1. Introduction

Fungi are ubiquitous in nature, having the capacity to colonise a wide range of ecosystems, including crops and foodstuffs worldwide. Several studies have reported the potential of these organisms in the production of useful organic compounds with numerous pharmaceutical benefits such as antibiotics, e.g., penicillin [1]. However, toxigenic fungi, particularly those belonging to the genera *Aspergillus, Fusarium, Penicillium, Alternaria*, and *Claviceps* are of importance because of their ability to produce several toxic secondary metabolites (mycotoxins) under favourable environmental and ecological conditions [2,3]. The production of these mycotoxins, though not necessarily important for the fungal growth, is considered to serve as a defence against predators and change in environmental conditions in fungi’s ecological niche [1,4,5]. Mycotoxins are a chemically diverse group of low molecular compounds, which occur in agricultural commodities and processed food products, as well as in the environment. To date, 300 to 400 mycotoxins are known, with those of special interest being aflatoxins (AFs), fumonisins (FBs), ochratoxins (OTs), trichothecenes (THs), zearalenone (ZEN), citrinin (CIT), and *Alternaria* toxins. The importance placed on these toxins may be linked to their principal roles in occurrence, distribution, and toxicity.

The contamination of food products by mycotoxins can occur all through the life and process cycle of crops, starting from the pre-harvest, harvest, and storage stages, as well as the process stage. Moreover, certain fungi and types of toxin production are dominant at specific stages. While pre-harvest stages favour the growth of species of *Fusarium, Alternaria,* and *Claviceps* and associated mycotoxin production, *Aspergillus* and *Penicillium* species strive maximally during storage [6,7]. The occurrence of mycotoxins worldwide is unavoidable; however, the concentration at which these toxins occur in food products is of great concern. Mycotoxin occurrence has been reported in several agricultural crops and products, including cereals and cereal-based products, pulses and pulse-based products, fruits and fruit-based products, nuts, spices, coffee, and tea [8,9,10,11,12]. A great concern is the occurrence of this array of toxins in foods destined for infants and young children (IYC) and their possible presence in breast milk as a result of maternal exposure. Although constant efforts are being made to reduce fungal infestation and mycotoxin contamination in food products, human and environmental factors such as climatic change, pest activities, and improper agricultural and storage practices hamper these efforts. In addition to the economic cost on crops, which is worth billions of dollars, mycotoxins have the potentials to induce acute or chronic adverse health consequences, known as mycotoxicoses, in humans and animals.

Exposure to these toxins can be through ingestion, inhalation, or/and dermal absorption. The degree of toxic effect is dependent on the toxin type, exposure dose and duration, age, sex, and health status of the host, exposure route, and possible synergistic effects of other chemicals to which the individual is exposed [13,14,15,16,17]. The potential subgroup most affected by mycotoxin contamination is IYC, probably because of their high ingestion of mycotoxins due to frequent consumption of cereal-based food in proportion to their body weight [18,19]. Additionally, the young developing organs and immune systems of IYC may predispose them to the toxic effect. Mycotoxin exposure has been linked to a wide range of diseases ranging from mild symptoms of nausea, vomiting, and dizziness to long-term degenerative diseases as a result of carcinogenic, nephrotoxic, neurotoxic, hepatotoxic, genotoxic, immunotoxic, estrogenic, and teratogenic properties [20,21,22]. In the case of children, mycotoxins have been associated with developmental defects, such as neuro-developmental disorders. Recent epidemiological studies by De Santis et al. reported a significant association between mycotoxins and autism spectrum disorder (ASD) [23,24]. A high incidence of neural tube defects (NTDs) in the areas of the world, such as African communities, where maize serves as a major staple food has been linked to FB contamination in maize [25,26].

Another study also reported the capability of mycotoxins such as AFs to cross the placental barrier, thus causing alteration of foetal health [27,28]. Their implication with pulmonary hemosiderosis in infants, resulting in anaemia, chronic cough, dyspnea, wheezing, and often cyanosis, has also been reported [29]. Other health effects, such as undernutrition and increased morbidity and mortality of infants as a result of chronic aflatoxicosis, resulting in immune malfunction and poor absorption of micronutrients, has also been recorded [30]. A study in 1989 implicated mycotoxins, specifically ZEN and its metabolites, to have an estrogenic effect, thus resulting in premature breast development and precocious sexual development in children in Puerto Rico between 1978 and 1981 [31,32]. In addition, exposure to toxigenic fungi and their metabolites have been linked to child growth and weight impairment [6,33,34,35,36,37,38,39,40,41]. This scenario of stunting and wasting is often seen in the developing continents such as Africa [42], especially in SSA. This region is characterised by the frequent occurrence and high levels of contaminants such as mycotoxins in food products and food for infants and young children, including complementary and weaning foods, as well as breast milk [9,38,43,44,45,46].

Stringent regulatory limits for mycotoxins have been established by countries, especially in the developed world, thus translating into reasonable protection for consumers, specifically IYC. However, developing regions, such as SSA, hitherto still face outbreaks of mycotoxicosis directly or indirectly, which are often unreported. Of most concern is the attitude of the region towards protecting the most vulnerable group, which is evident in the non-establishment and enforcement of a collective regulatory law for the region. In 2004, the Food and Agricultural Organization (FAO) reported the outcome of a worldwide survey, which identified only 15 African countries to have mycotoxin regulation, out of which 11 are situated in SSA [47]. Unfortunately, to date, there is little or no improvement in the region with respect to mycotoxin regulation, with very weak enforcement power in the countries with existing laws. The shortfalls experienced in this region may be attributed to food security status which impedes food safety efforts within the food value chain [48]. The non-proactiveness of the food safety bodies and the complex agricultural farming system in the region further complicate the situation. Another possibility is inadequate scientific data, including occurrence, exposure, and risk assessment data in SSA. This necessitates the urgent need for a comprehensive review to summarise existing published data regarding the occurrence of mycotoxins and its associated risk in SSA, particularly with reference to IYC.

Therefore, in the current review, we present an overview of the data on the occurrence of mycotoxins in IYC foods, including complementary foods, as well as breast milk, in SSA. Furthermore, it summarises the previous studies on mycotoxin exposure of IYC as well as health implications of mycotoxins in IYC with particular reference to the SSA region.

## 2. Mycotoxin Occurrence in Foods for Infants and Young Children

The complex and ubiquitous nature of mycotoxins makes them occur virtually in every environment. There is ambient evidence for their occurrence in human foods, animal feeds, and environmental air [49,50,51,52,53]. While there is a wealth of data on the occurrence of mycotoxins in infant foods in developed countries, it will become evident from this review that only a few studies are available on this subject in SSA. Therefore, it is important to assess the extent to which foods intended for the most vulnerable populace group (IYC) in SSA are contaminated with these toxins.

### 2.1. Cereal-Based Products

Cereals are ranked the major susceptible crops to mycotoxin contamination, with maize being at the forefront [54]. Because of their excellent energy and nutritional sources, these crops serve as the main ingredient used in the production of foods for IYC [55]. Although studies have shown that the processing of food can lead to a reduction or degradation of mycotoxins, it is important to highlight that mycotoxins can be carried over to the final processed products, thus necessitating the need for good quality raw material [56,57,58]. Available studies in the world have revealed the occurrence of both regulated and non-regulated mycotoxins in foods for IYC, including the so-called “emerging mycotoxins” [59,60,61,62,63,64,65,66,67,68,69,70,71]. In SSA, only a few studies have focused on food products for infants and children. Notwithstanding the few available data, the prevalence of mycotoxins in IYC foods is obvious.

Mycotoxins have been found at high concentrations in foods for IYC in SSA countries, such as Nigeria, Tanzania, and Burkina Faso, with most levels exceeding the regulatory limits set by the EU and some SSA countries (Table 1) [9,38,72,73]. Kimanya et al. [38] reported the frequency of FBs in ready-to-cook maize flour consumed by infants in the division of Tarakea, Tanzania. Of the 191 samples examined in their study, the authors observed that 131 of the samples were contaminated with FBs levels, ranging from 21 to 3201 µg/kg. The same group of authors examined maize flour consumed by 41 children in the rural village of Kikelelwa, Tanzania and detected multiple mycotoxins, including AFs (range, 0.11–386 µg/kg), deoxynivalenol (DON) (range, 57–825 µg/kg), and FBs (63–2284 µg/kg) at 32%, 44%, and 83%, respectively [72]. A similar trend was observed in other studies carried out in other countries of SSA. A Nigerian study reported a high occurrence rate (97%) of *Fusarium* mycotoxins in traditional cereal-based infant foods (*ogi*) sampled from local markets, with one of the samples having a total FB level as high as 3557 µg/kg. Another study on *ogi* from Nigerian and South African markets also reported the high tendencies of mycotoxin contamination in the product [43]. 

In a more recent study, the mycotoxicological contamination of 137 industrially processed and household formulated complementary food samples fed to Nigerian IYC was assessed [74]. Out of the mycotoxins quantified, 29 were detected in 84 cereal- and nut-based complementary food samples. A total of 27 and 16 mycotoxins were detected in the industrially processed products—tom bran (mixed grains containing peanut) and family cereal had a total AF frequency of 83.3% and 100%, respectively. As for the traditionally processed product (*ogi*), 19 toxins were detected, with a 52.2% occurrence rate of total AFs (Table 1). In addition, the authors recorded the occurrence of other EU-regulated mycotoxins in the cereal-based foods, including FBs, ZEN, ochratoxin A (OTA), and DON. These mycotoxins were also reported in a more recent study from the same country [75]. Similarly, Ware et al. [73] investigated the occurrence of AFs, OTA, and FBs in 199 infant formulas and 49 cereal- and oilseed-based infant products marketed and consumed in the capital city of Burkina Faso using high-performance liquid chromatography/fluorescence detector (HPLC/FD). The study revealed a 73.4% (182/248) occurrence rate of mycotoxins in the samples. The authors further highlighted that out of the 17 samples of maize and rice analysed, 23.5% and 17.7% exceeded the EU regulatory limits for AFB1 and AFs, respectively. Meanwhile, for peanuts, cereals and cereal–based products other than maize and rice, 39.3% and 35.7% of the samples were above EU limits for AFB1 and AFs, respectively.

### 2.2. Breast Milk and Infant Formula

Although the literature has recognised the numerous benefits of breast milk in the development and health of infants [79], it has also been established that breast milk may serve as a route of exposure of infants to environmental toxins such as mycotoxins. The contamination of breast milk is highly related to the maternal dietary habits through the consumption of contaminated foods, which is highly influenced by the socio-demographic status of the mother and seasonal variations. While it is evident that the physicochemical properties of the toxic compounds, as well as the biochemical characteristics of milk, such as high lipid content and low pH, contribute to the excretion of toxic substances including mycotoxins into milk [80], there are still major knowledge gaps on the uptake and lactational transfer of mycotoxins to human breast milk. Moreover, there is evidence to suggest that the transfer of mycotoxins into breast milk may also depend on the frequency of infant feeding as well as the occurrence of breast infections as a result of breast milk production and feeding [80,81].

Several studies, especially from the EU, have demonstrated the occurrence of mycotoxins and their metabolites in breast milk [81,82,83,84,85,86,87,88,89,90]. While aflatoxin M1 (AFM1) and OTA are the most studied mycotoxins in breast milk, OTA seems to be the major mycotoxin detected in human breast milk from EU regions [88,89]. In contrast, AFB1 and its metabolite AFM1 dominate human breast milk samples from SSA (Table 2), with the majority exceeding the EU regulatory limits set for processed infant/child foods [91,92,93,94]. Alegbe et al. [92] and Adejumo et al. [91] reported high occurrence rates of 82% of AFM1 in breast milk samples from Nigeria, with the highest concentrations being 70 ng/L and 35 ng/L, respectively. A much lower frequency was observed in an earlier study from Nigeria using thin layer chromatography (TLC) which reported AFM1 contamination of five (18%) out of 28 breast milk samples [95]. The result reported by these authors may have been influenced by the sensitivity of the analytical method used. Kang’ethe et al. [96] performed a comprehensive study on the occurrence of AFM1 in breast milk from two counties (Makueni and Nandi) in Kenya using two different analytical methods (enzyme-linked immunosorbent assay (ELISA) and HPLC) and observed the occurrence of AFM1 in the samples irrespective of the methods used. A high frequency of AFM1 contamination of 87% and 57% out of 98 and 67 samples, respectively, and a concentration range of 0.23–48 ng/L (Makueni) and 0.003–3.7 ng/L (Nandi), were registered using the ELISA method. When using the HPLC method, 22.2% (range: 1.4–153 ng/L) and 9.5% (range: 0.5–0.8 ng/L) out of 18 and 21 samples from Makueni and Nandi, respectively, were contaminated with AFM1.

The frequency of AFM1 in breast milk samples was also reported in other countries in SSA, including Tanzania, Cameroon, Kenya, and Sudan [45,93,94,99]. In addition to AFM1, a Tanzanian study using HPLC with a fluorescence detector reported FB1 in 44.3% of 131 breast milk samples. This was succeeded by a confirmation study using a high-resolution liquid chromatographic separation technique to validate the FB1 levels in the samples. According to the study, the concentration of FB1 ranged from 6570 ng/L to 471,500 ng/L, with about 10.3% of the samples exceeding the FB1 regulatory limits of 200 µg/kg set by the EU [93]. Another study using a highly sensitive and specific multi-mycotoxin assay reported the occurrence of OTA, beauvericin (BEA), enniatin B (ENN B), and AFM1 in breast milk samples from Nigeria out of the 28 mycotoxins evaluated [90]. Interestingly, BEA was the dominant toxin, contaminating 56% of the samples with a concentration up to 19 ng/L. Ochratoxin A, ENN B, and AFM1 were detected in 15%, 9%, and 1% of the samples, respectively, with the concentration being below the limit of quantification (LOQ), except for one sample with ENN B at a concentration of 9 ng/L [90]. This is in line with a recent study by Ezekial et al. [75], which detected nine mycotoxins, including AFM1, alternariol methyl ether (AME), BEA, dihydrocitrinone (DCIT), ENN B, ENN B1, OTA, ochratoxin B (OTB), and sterigmatocystin (STG) in breast milk samples from Nigeria. Another study also reported a 40% co-occurrence rate of AFs and OTA in breast milk samples obtained from 113 mothers from Sierra Leone, with the individual toxin concentrations ranging from 3-372,000 ng/L for AFs (AFB1, AFM1, AFM2, AFG1, AFG2, and aflatoxicol) and 200–337,000 ng/L for OTA [98].

According to the US Food and Drug Administration, infant formula, which represents a special diet, is used solely as infant food by reason of its simulation of human milk or suitability as a complete or partial substitute for human milk (FDA, 2018). Although infant formula serves as complementary food for IYC and supplies their daily nutritional requirements, these products may also be a route of mycotoxin exposure to this age group. The occurrence of mycotoxins in infant formulas has been reported in different parts of the world. Meucci et al. (2010) investigated the occurrence of AFB1 and OTA in 14 leading brands of infant formulas available in the Italian market. While 133 (72%) samples were positive of OTA (range: limit of detection (LOD)–0.690 µg/kg), only two (1%) samples were positive of AFM1 (range: LOD–0.015 µg/kg) [101]. Another study on the occurrence of mycotoxins in infant formula marketed in Ankara, Turkey using ELISA detected AFB1, AFM1, and OTA at a rate of 87%, 36.5%, and 40%, and within a range of 0.10–6.04 µg/kg, 0.06–0.32 µg/kg, and 0.27–4.50 µg/kg, respectively [102]. Other countries, such as Pakistan, Canada, Brazil, and South Korea, have also reported mycotoxins in infant formulas [103,104,105,106].

A few studies from SSA also revealed a high occurrence of various mycotoxins in infant formula (Table 2). Ware et al. reported mycotoxin contamination of infant formulas sampled from the city of Ouagadougou, Burkina Faso [73]. AFB1 was the most dominant mycotoxin, occurring at a rate of 83.9% with a concentration of up to 87.4 µg/kg. Other toxins detected included OTA and FBs (FB1 + FB2), with 7.5% and 1.5% of the samples exceeding the EU regulatory limits and having concentrations up to 3.2 µg/kg and 672.9 µg/kg, respectively. This study suggests that IYC in this region may be highly exposed to mycotoxins, considering the high occurrence rate and maximum concentrations, which were 900, six, and three times higher than the EU regulatory limits of 0.1 µg/kg (AFB1), 0.5 µg/kg (OTA), and 200 µg/kg (FB) set for foods for IYC, respectively. Equally, Ojuri et al. [74] investigated the distribution of mycotoxins in 17 infant formula samples routinely fed to IYC in Nigeria and observed multiple occurrences of mycotoxins. ZEN was the most prevalent toxin, occurring at a rate of 23.5% (range: 0.4–5.4 µg/kg), followed by BEA (17.5%, range 0.1–13.4 µg/kg). Trichothecenes (DON, nivalenol (NIV), and T-2 toxin (T-2)) and moniliformin (MON) were detected at a frequency of 11.8%, while AFB1 and AFB2 were detected in one sample at a concentration of 4.2 µg/kg and 0.5 µg/kg, respectively.

### 2.3. Oilseed- and Fruit-Based Products

In addition to cereals, the utilisation of other plant products such as pulses and oilseeds, fruits, and vegetables in the production of IYC foods has been reported [107,108,109]. Their uses have been encouraged, especially in developing regions, such as SSA, to help combat the high rate of malnutrition and its associated health disorders in the region. Soybean and groundnut serve as cheaper alternatives in the absence of animal protein. However, studies have shown that IYC food products processed with these plant products may also be contaminated with mycotoxins [70,110]. A Nigerian study reported the occurrence of *Fusarium* mycotoxins in processed soybean powder, a major complementary weaning food in the country [48]. The result revealed the frequency of ZEN (81%), 15-acetyldeoxynivalenol (15-AcDON) (31%), FBs (28%), and HT-2 toxin (HT-2) (25%) in the samples analysed. AF, FB1, and OTA have also been detected in processed soybean powder samples from Nigeria at incidence rates of 45, 100, and 40%, respectively [111]. On the other hand, the susceptibility of groundnut/peanut to fungi and their metabolites has been reported in several countries in SSA [74,77].

Peanut butter is used in weaning foods such as porridges. A study conducted by Ojuri et al. [74] on five samples of peanut butter from Nigeria revealed the occurrence of mycotoxins in 80% of the samples, including AFs (80%), BEA (80%), MON (60%), and 3-nitropropionic acid (20%) in ranges of 6.5–13.6 µg/kg, 0.6–4.1 µg/kg, 2.2–3.5 µg/kg, and 5.4 µg/kg, respectively (Table 1). Another group of authors conducted a comprehensive multi-year survey of AF contamination in 954 containers of 24 local and imported peanut butter brands collected between 2012 and 2014 from Zambian shops [77]. Overall, 73%, 80%, and 53% of the 24 peanut butter brands analysed in 2012, 2013, and 2014 were contaminated with AFs, with AFB1 concentrations of up to 130 µg/kg, 10,740 µg/kg, and 1000 µg/kg, respectively. It is important to note that some of the brands analysed in this study were manufactured and imported from South Africa, Malawi, and Zimbabwe, thus suggesting that IYC in these countries may also be predisposed to these toxins. The result observed by these authors is in line with the studies that reported the occurrence of AFs in peanut butter and groundnut products from Zimbabwe and Zambia, respectively [78,112].

Mycotoxins may also contaminate fruit and vegetables, and subsequently be transferred into their processed products because of their chemically stable nature, even when exposed to high temperatures. Studies from EU countries like Germany, Spain, Portugal, and Serbia, and other parts of the world, have reported the occurrence of mycotoxins, especially patulin (PAT), in fruit-based infant food products [63,113,114,115,116]. Whilst there are data available on mycotoxin contamination of fruit- and vegetable-based IYC foods in different regions of the world, unfortunately, there is little or no information in SSA, which suggests the need for more studies in the region to ascertain the extent of mycotoxin contamination of these products.

## 3. Mycotoxin Exposure Assessment of Infants and Young Children in Sub-Saharan Africa

In recent years, extensive attention has been channelled to the most vulnerable group regarding the toxic effects exerted by mycotoxins. This is evident in the continuous monitoring of foods for IYC, assessment of exposure, and characterisation of risk associated with IYC exposure to mycotoxins and the stringent establishment and enforcement of regulatory limits to protect this subgroup of the population. While these efforts are functional in developed countries, economically developing regions such as SSA still face the challenge of high levels of mycotoxin exposure of IYC due to the endemic nature of food contamination in the region. To this effect, only a few studies in SSA have evaluated and reported the exposure of IYC to mycotoxins, as well as the associated risk.

### 3.1. Dietary Exposure Assessment

The estimation of dietary exposure can be based on either a numerical point estimation (deterministic) approach or a stochastic (probabilistic) approach, which uses a random probability distribution pattern. Ojuri et al. [74] assessed the mycotoxin (AFs, FBs, OTA, CIT, BEA, and MON) exposure of Nigerian IYC fed with complementary foods using the deterministic approach. The authors obtained information on the daily consumption rate of complementary food using a structured questionnaire, while mycotoxin occurrence and concentrations of the foods consumed by the IYC population were determined by LC-MS/MS. The study revealed high exposure estimates for several mycotoxins (Table 3). The authors observed that total AF dietary exposure ranged from 26–54,892 ng/kg bw per day, thus suggesting an AF exposure rate much higher than the established toxicological reference point. In the case of total FBs, maximum exposures were about 69-fold higher than the tolerable daily intake (TDI) (2µg/kg bw per day) recommended by the Joint FAO/WHO Expert Committee on Food Additives (JECFA) [117] with an estimated exposure range of 0.0–138.6 µg/kg bw per day [74]. Similarly, OTA exposure for IYC ranged from 0.01 µg/kg bw per week (upper bound value) to 2.03 µg/kg bw per week, which is up to 20 times more than the tolerable weekly intake (TWI) of 0.1 µg/kg bw per week for OTA, as recommended by the JECFA [118]. For CIT, a chronic exposure ranging from 0.002–102 µg/kg bw per day was reported by the authors, considerably higher than the TDI of 0.2 µg/kg bw per day recommended by EFSA [119] at which there are no nephrotoxicity concerns in humans. This study suggests that Nigerian IYC may be exposed to high levels of toxic secondary metabolites on a daily basis and implies a major health concern. The same trend was observed by an earlier study, which assessed the exposure of IYC to mycotoxins through the consumption of maize grains as complementary foods in Nigeria using the probable daily intake (PDI) approach [120].

Maize is an important cereal used for both traditional and industrial processing of infant complementary foods in SSA [57]. By modelling Tanzanian maize consumption data with previously reported FB occurrence data for both sorted and unsorted maize harvested in 2005 and 2006, Kimanya et al. [121] reported that 26% and 3% of the infants exceeded the provisional maximum tolerable daily intake (PMTDI) of 2 µg/kg for FBs set by the JECFA, respectively. The 50th and 97th percentiles of maize consumers in 2005 were exposed to FB mean levels of 0.47 and 36.99 µg/kg bw/day, respectively. However, the authors observed a lower exposure rate of 0.15 µg/kg bw/day (50th percentile) and 2.06 µg/kg bw/day (97th percentile) in 2006 [121]. The consumption of unsorted maize led to a significantly higher FB exposure level (8,87 µg/kg bw/day) in IYC when compared to an FB value of 0.28 µg/kg bw/day estimated for IYC fed with sorted maize. While studies have highlighted the high mycotoxin exposure of fully weaned (complementary foods) infants compared to breastfed infants [34,75], evidence of exposure of breastfed infants to mycotoxins has also been reported. Magoha et al. [93,100] reported mycotoxin exposure data of infants consuming breast milk in Tanzania. These authors revealed infant exposure to FBs at levels ranging between 0.78 and 64.93 µg/kg bw/day as a result of the consumption of breast milk. Among the studied group, 29% of the infants exceeded the provisional maximum tolerable limit (2 µg/kg) [93]. This trend of infant exposure due to the consumption of breast milk was also observed when other mycotoxins were assessed [100].

In addition, it is important to note that co-exposure to multiple mycotoxins may occur, which could lead to either synergistic or/and additive effects on the host, though toxicity is dependent on exposure time and dose as well as animal species. A study estimated the co-exposure of rural Tanzanian children fed with maize-based complementary foods to multiple mycotoxins, including AFs, DON, and FBs [72]. In the study, out of the 41 children evaluated, 41% were exposed to both DON and FBs, while 29% of the children were exposed to AFs and FBs. Co-exposure of the three mycotoxins assessed (AFs, DON, and FBs) was observed in 10% of the children. For individual toxins, 32% out of the 41 children were exposed to AFs at exposure levels of 1–786 ng/kg bw/day, thus exceeding the AF exposure limit (0.017 ng/kg bw/day). Forty-four percent and 83% of the children had DON and FB exposure, ranging from 0.3–8.87 µg/kg bw/day and 0.19–26.37 µg/kg bw/day, respectively. Among the FB- and DON-exposed children, 56% and 66% exceeded the provisional tolerable intakes of 2 µg/kg bw/day and 1 µg/kg bw/day, respectively [72]. When compared with an earlier exposure study by Kimanya et al. [38] from the same country, the researchers observed 15% higher FB exposure rates with 46% of the children exceeding PMTDI (2 µg/kg bw/day) of FB. They concluded that the variation may be due to the fact that the study focused on older children who probably consumed more foods compared to the infant cohort used in their previous study.

### 3.2. Mycotoxin Exposure Assessment Using Biomarkers

Beside estimating exposure based on food consumption and mycotoxin concentration in food, the assessment of mycotoxin exposure using biomarker analysis has gained prominence in recent years. Biomarker analysis covers mycotoxin intake from all dietary sources and exposure routes [122], unlike the use of the occurrence and dietary intake approach which, due to heterogeneous distribution of mycotoxins in food products and limited accuracy in consumption data, often lead to under- or overestimation of exposure and, consequently, risk. Mycotoxin biomarker analysis in SSA is still in its infancy, albeit available exposure studies using the approach have shown consistently high levels of exposure in IYC (Table 4) [39]. Mycotoxins such as AFs are lipophilic in nature, have the ability to cross the placental barrier, and can be bioactivated in utero. It is worrying that a huge number of infants in SSA are pre-exposed to mycotoxins early in the uterus through a transplacental pathway as a result of maternal exposure, thus resulting in developmental issues [36,123,124,125].

Gong et al. [34], using serum AF-albumin adducts, reported the high AF exposure of children in Benin, West Africa, with AF-albumin levels as high as >1100 pg AF-lysine equivalents per milligram of albumin. There were strong variations in the levels of AF exposure depending on the village and geographical region. This is possibly linked to the climatic conditions (temperature and humidity) of the environment, which favours the growth of fungi and subsequently mycotoxin production. The pre- and post-agricultural practices, which differ from region to region, may also be a contributory factor [135]. This is evident in the variation of AF-albumin prevalence (February (98%), June (99.5%), and October (100%)) observed by the authors as a result of seasonal changes, even though the variation was not significant [34]. Earlier studies by this group of authors observed a remarkable increase in IYC exposure to AFs following the introduction of and increase in the consumption of weaning foods [34,126]. Similar results were reported in a South African study on non-breastfed infants [136]. Significant levels of OTA were detected in the plasma of non-breastfed infants at 6 weeks old as compared to breastfed infants. The authors also highlighted a progressive trend in the exposure up until the 8th week. Further studies originating from other countries in SSA using serum and plasma biomarkers have also indicated the exposure of IYC to AFs and FBs [33,36,126,127,128,129,137].

Consistent with these data, exposure studies using urinary biomarker approaches have indicated high concentrations of mycotoxins in IYC [128,130,131,132,133,134,137,138]. A Cameroonian study investigated the mycotoxin exposure of 220 children between the age of 1.5–4.5 years using urinary biomarkers [130]. Of the children investigated, mycotoxins were detected in 73% of the urine samples, with OTA being the most prevalent (30%) toxin, occurring within a concentration range of 0.04–2.4 ng/mL. Other mycotoxins detected in the urine samples included FB1 (11%), DON (17%), AFM1 (14%), ZEN (4%), α-zearalenol (α-ZEL) (4%), and β -zearalenol (β-ZEL) (8%), with up to two, three, and four mycotoxins co-occurring in samples at a rate of 35%, 5%, and 5%, respectively [130]. The detection of multiple mycotoxins in IYC urine has also been reported by other studies from the region. Ezekiel et al. [133] detected FB1, AFM1, OTA, and deoxynivalenol-15-O-glucuronide in the urine samples of Nigerian children, with a mycotoxin prevalence rate of 47.3% (9/19). Using an ultra-sensitive stable isotope-assisted quantification method, the same group of authors further confirmed the exposure of Nigerian IYC to multiple mycotoxins [138].

Ayelign et al. [131] indicated the presence of AFs in 17% of urine samples of Ethiopian children, with AFM1 being the most prevalent (7%) toxin. Other AFs were detected in 4.5% (AFB2), 2.5% (AFG1), and 3% (AFG2) of samples. A similar study in Guinea reported the occurrence of AFs in children’s urine samples at a higher occurrence rate of 86% [132]. While AFB1, AFB2, AFG1, and AFG2 were detected in the samples, AFM1 was the dominant mycotoxin, occurring at a frequency and concentration range of 64% and 8.0–801 pg/mL, respectively. Using a urinary biomarker, Shirima et al. [128,137] revealed high levels of FBs among Tanzanian IYC, with individual levels as high as 698.2 ng/L. The authors further observed a discrepancy in mycotoxin exposure levels between the sampling times, which could be linked to the effects of consumption of contaminated food as well as seasonal variation of mycotoxin contamination levels [128]. The same trend was observed for DON among IYC in Tanzania, with concentration ranges of 800–1400 ng/L, 1700–3200 ng/L, and 4100–7900 ng/L at recruitment, after 6 months, and after 12 months, respectively [134].

## 4. Health Implications Associated with Mycotoxin Exposure of Infants and Young Children in Sub-Saharan Africa

The health significance of human exposure to mycotoxins should not be neglected, especially with regard to infants and young children in SSA. This population group on a daily basis is subjected to toxic metabolites through food consumption, inhalation, and dermal adsorption. Albeit there are no studies on the exposure to and effect of mycotoxins as a result of inhalation and dermal adsorption in SSA, there are several reports from this region on the detrimental effects of ingestion of mycotoxins, especially AFs and FBs, on the health of IYC.

### 4.1. Child Growth Impairment

Researchers have studied the possible association between mycotoxin exposure, especially to AFs and FBs, and IYC development. The few studies conducted to determine the relationship between mycotoxins (AFs and FBs) and IYC growth have been reviewed by Smith et al. [42], Gong et al. [139], and Tesfamariam et al. [140].

#### 4.1.1. Aflatoxin Exposure and Infant Growth

Using a cross-sectional study design, Okoth and Ohingo [37] investigated the possible association between child dietary exposure to AFs and growth impairment among 242 children within the age range of 3 to 36 months from Kisumu District of Kenya. They revealed a highly significant relationship between the number of children fed with mycotoxin-contaminated flour and the prevalence of wasting (*p* = 0.002) (Table 5). In another study in Kenya with 204 children (age: 1–3 years), AFM1 exposure was negatively associated with height-for-age Z-score (HAZ) score (*p*=0.047), however, there were no associations between total AFs and HAZ, weight-for-age Z-score (WAZ), or weight-for-height Z-score scores (WHZ) [44]. The authors found prevalence of stunting, underweight, and wasting of 34%, 30%, and 6%, respectively, of which 53.8% of the children classified as wasted consumed AF-contaminated flour compared with 27.7% of not wasted children (*p*=0.002). Sixty percent of the 30.7% of severe protein energy malnourished children were weaned with AF-contaminated flour compared to 27.4% of the normal children (*p* = 0.004).

A similar study in Nigeria reported a significantly higher concentration of AF-lysine in stunted and severe acute malnourished children compared to normal children [143]. This is in agreement with the study by Hoffmann et al. [142] that examined the effectiveness of reducing AF exposure on child linear growth and AF serum levels of infants and young children in rural communities in Eastern Kenya using a cluster randomised longitudinal study design. The study revealed a substantial reduction of endline serum AFs due to the consumption of AF-free maize, however, no effect on child linear growth of the examined population was observed. Furthermore, the midline analysis suggested the possibility of AFs having an effect on linear growth at younger ages [142].

A cross-sectional study by Gong et al. [33] examined the relationship between AF exposure and growth in IYC between the ages of 9 months and 5 years from four geographic zones of Benin and Togo. Using anthropometric data and AF-albumin concentrations according to the World Health Organization (WHO) Z-score criteria, the authors observed higher AF-albumin in children with stunting, underweight, and wasting with a prevalence rate of 33%, 29%, and 6%, respectively. Notwithstanding that the AF-albumin concentration negatively correlated with the growth parameters, they were highly significant [33]. This observation was supported by a later longitudinal study in the Republic of Benin, which after adjustment for socioeconomic status, anthropometric parameters, agro-ecological zone, and weaning status, showed a negative correlation between AF-albumin and height increase over the 8-month follow-up period, hence suggesting a strong association between AFs and stunting [34]. The same trend was recorded by Watson et al. [141], who reported an inverse relationship between AF-albumin adducts and HAZ, WAZ, and WHZ scores in a cohort of 374 Gambian infants between the ages of 0 and 2 years.

Lauer et al. [146] investigated the association between maternal AF exposure during pregnancy and birth outcomes in Uganda. The authorsobserved significant associations between maternal AFB-lysine levels and lower birth weight, lower WAZ, smaller head circumference (HC), and lower head circumference-for-age Z-score (HCZ) at birth. Conversely, no significant associations were observed between maternal AFB-lysine levels and infant length, WHZ, HAZ, or gestational age at birth [146]. Another longitudinal study from Gambia examining the effect of in utero AF exposure on infant growth using 138 pregnant women and their infants reported a significant association between maternal AF exposure during pregnancy and growth faltering in infants [36]. The same trend was observed in a similar study on 143 lactating mothers and their infants in Tanzania, with a significant (p<0.05) inverse association between AFM1 exposure and stunting (HAZ score) and underweight (WAZ score) in infants fed with AFM1-contaminated breast milk [100]. In Uganda, using 246 dyads, Natamba et al. [145] investigated the association of perinatal exposure to AFs with a low rate of weight gain among human immunodeficiency virus (HIV)-positive pregnant women and reduced linear growth of HIV-exposed infants. They found a negative effect of AF exposure on infant linear growth in HIV-positive pregnant women and their infants, with the infants of HIV-positive women in the high perinatal AF category having a lower HAZ scores (0.460) when compared with the infants of HIV-negative low-AF-exposed women (p=0.006) [145]. Shirima et al. [128] also conducted a similar longitudinal study in three agro-ecological zones of Tanzania, investigating the effects of AF exposure and co-exposure on the growth of IYC using a cohort of 166 apparently healthy IYC (aged 6 to 14 months old). The study revealed no significant association between AFs and child growth irrespective of the sampling time, which the authors attributed to the lower AF-albumin concentrations (geometric mean: 4.7 pg/mg) recorded. This is in line with an Ethiopian study [131], but in contrast to earlier studies reported in West Africa [33,34,36], Tanzania [100], and Kenya [37]. Furthermore, a non-significant association between AFs and infant growth was later reported by a study in Haydom, Tanzania, using a cohort of 114 children under the age of 36 months [41], which is in line with other studies from other regions of the world [147]. In addition to the variation in the concentrations of AFs, it is important to highlight that the different methodologies used in these studies may have influenced the variation in the effects observed.

#### 4.1.2. Fumonisin Exposure and Infant Growth

Shirima et al. [128] observed that the concentrations of urinary FBs (0.31, 0.17, and 0.57 ng/mL) detected in a cohort of 166 healthy children were negatively associated with growth impairment (rate: 44%, 55%, and 56%) at recruitment, 6 months, and 12 months, respectively, suggesting a possible association between FBs and child growth (Table 5). Chen et al. [41] also reported a negative association of FBs with stunting, though at a higher incidence rate of 75%, which is attributed to the higher urinary FB concentration (mean: 1.3 ng/mL) recorded in the study when compared to Shirima et al. [128]. This therefore implies a possible dose-dependent effect between FBs and stunting (increase in FB concentration = increase in stunting) [41]. An earlier study originating from the same country reported a similar trend [38]. The authors examined the effect of FB exposure on the growth performance of 215 infants complemented with maize-based complementary foods. They revealed that infants exposed to FBs above the PMTDI of 2 µg/kg body weight/day established by the JECFA were significantly shorter (1.3 cm) as compared to the other infants [38]. Another three-time point (1, 3, and 5 months) study investigating the relationship between FB exposure and stunting or underweight in 143 Tanzanian infants under 6 months of age showed insignificant associations between the variables [76]. Furthermore, it is important to mention that the influence of mycotoxins on infant growth is not only an SSA problem, as studies originating from other regions of the world have reported strong correlations between mycotoxin exposure and infant growth impairment [148,149,150].

#### 4.1.3. Postulated Mechanism for Growth Impairment

Although the mechanism by which mycotoxins influence child growth is yet to be fully understood, Smith et al. [151] suggested two possible pathways by which these toxins may contribute to stunting, including the induction of environmental enteric dysfunction and systemic immune activation, resulting in the interruption of the insulin-like growth factor 1 (IGF1) axis. Castelino et al. [152] investigated the relationship between AF and IGF1 in 199 Kenyan schoolchildren, and observed that AF-albumin concentrations were inversely associated with both IGF1 and IGF-binding protein-3 (IGFBP3). IGF facilitates most of the growth-promoting effects of growth hormone, and thus plays a major role in the growth of a child [153]. AFs may also induce stunting by exerting an immunosuppressive effect, which often increases the susceptibility of infants to infection, loss of appetite, and reduced nutrient absorption [154]. In addition, AFs may exert infant growth impairment by mediating intestinal damage through the inhibition of protein synthesis [151]. Other mycotoxins like DON have also been postulated to exert growth impairment through this mechanism.

On the other hand, FBs may contribute to impaired growth in infants by inhibiting ceramide synthase, a major enzyme responsible for the biosynthesis of sphingolipids, hence leading to the disruption of sphingolipid metabolism [155]. This is in line with the study by Semba et al. [156], who reported significantly lower serum concentrations of sphingomyelins in stunted children when compared with non-stunted children in six villages in rural southern Malawi. Sphingomyelin, localised in the plasma membrane, is a dominant sphingolipid in the membranes of mammalian cells, and plays a major role in creating lateral structures in membranes for Toll-like receptors and class A and B scavenger receptors, as well as insulin receptors [157]. Wu [158] also reported the involvement of sphingomyelin in cell signalling. The influence of FBs on intestinal barrier function by the alteration of the sphingoid base-1 phosphate signalling pathway has also been reported. Riley et al. [159] revealed a positive correlation between the concentrations of urinary FB1 and sphinganine 1-phosphate/sphingosine 1-phosphate and sphinganine 1-phosphate ratio in blood. The result was further confirmed by a follow-up study involving 299 Guatemalan women. In addition to ceramide synthase inhibition by FBs, Masching et al. [160] reported the potential of FBs to induce inflammatory responses, thereby promoting damage to gut barrier function.

### 4.2. Child Immune and Nervous Systems

Exposure to mycotoxins has been shown to hamper immune responses in humans and animals, thus leading to a decrease in resistance to infectious diseases [35]. The mechanism through which mycotoxins exert immunosuppressive or immunostimulatory effects on humans and animals may vary depending on toxin type and exposure dose, as well as the investigated parameters highlighted in several reviews [161,162,163]. Bondy and Peska [162] highlighted the ability of mycotoxins to promote the expression of a diverse array of cytokines through inflammatory responses, with a potential to upregulate and downregulate a wide array of immune functions. While mycotoxins impair cell-mediated immunity and phagocytic cell function, exposure to AFs increases the T cell proliferation-inducing capacity of porcine monocyte-derived dendritic cells, therefore enhancing the antigen-presenting capacity of the cell [164]. The inhibition of humoral, cellular, and innate immunity by AFs, OTA, and FBs, leading to a reduced response to vaccines, has been reported in animal species [162,165].

Notwithstanding the evidence that IYC, especially in SSA, are most vulnerable to mycotoxin exposure, only a few scientific studies, mostly focusing on AFs, have been conducted to elucidate the effects of these fungus metabolites on the immunity of IYC. Allen et al. [166] investigated the relationship between AF exposure, hepatitis B infection, and the prevalence of malaria in 391 Gambian children between the age of 3 and 8 years, and found that a higher AF-albumin adduct concentration was associated with increased *Plasmodium falciparum* parasitaemia (*p* = 0.01) and hepatitis B surface antigen (p=0.04) carrier status of the children. However, the authors observed no consistent association between AF-albumin level, malaria infections, malaria-specific antibody, or lymphoproliferative responses. A later cross-sectional study from the same country, using a cohort of 472 children, studied the effect of dietary AF exposure on immune parameters, including secretory IgA in saliva (sIgA) and cell-mediated immunity, as well as antibody response to rabies and pneumococcal vaccines [35]. A remarkably lower sIgA level of children with detectable AF-albumin in their blood was seen compared to no AF-albumin detectable children (*p* < 0.0001). However, out of the four pneumococcal serotypes, only one response was weakly associated with higher levels of AF-albumin adducts. On the other hand, no association was observed between AF-albumin and cell-mediated immunity or rabies [35].

Other West African studies, specifically in Ghana, showed the potential effects of AF exposure on immune responses of HIV-positive participants [167,168,169,170], and tuberculosis patients [171], however, no such study on the infant population in these regions has been reported. Another study investigated the effect of OTA on T cell activation using a cohort of South African HIV-exposed infants and HIV-unexposed infants as controls [136]. The authors observed a correlation of OTA plasma levels with the activation of CD4 T cells (HLADR and CCR5) and a chemokine, CXCL10. Increased expression of CXCL10, HLA-DR, and CCR5 has been associated with a variety of human diseases, including HIV, immune dysfunction, chronic inflammation, infectious diseases, tumour development, and encephalopathy, known as a major cause of infant mortality in Africa [136,172,173,174], thus suggesting an increased risk of morbidity and mortality due to OTA exposure.

#### Mycotoxin Exposure and Autism Spectrum Disorder

Recent epidemiological studies have revealed possible relationships between mycotoxin exposure and neurodevelopmental disorders in IYC, especially as with regard to autism spectrum disorder (ASD) [23,24], a lifelong neuro-developmental syndrome characterised by deficits in social activities, communication interactions, and unusual restricted and repetitive behaviours. Scientific evidence indicates the role of multiple interacting genetic factors in the aetiology of ASD. With the recent increase in the incidence of ASD worldwide [175], at a rate of 60 cases per 10,000 children, there is a quest to identify more possible contributory factors to the disorder. Notwithstanding the attributable results of conceptual and technological advances in research, the possible role of environmental factors, such as toxic exposure, in triggering ASD should not be underestimated. In a cohort of 110 children, comprising 52 autistic and 58 healthy children (31 siblings and 27 unrelated subjects), De Santis et al. [23] examined the role of mycotoxin exposure in the manifestation of ASD. The authors reported a significant association between OTA concentrations in the serum (*p* = 0.0017) and urine (*p* = 0.0002) of the autistic children compared to the unrelated healthy children. In addition, a significant association was also found for OTA in urine when comparing ASD patients with healthy children (siblings and unrelated subjects) (*p* = 0.0081), whereas no significant difference was observed between ASD subjects and their siblings. The authors highlighted that the average values of mycotoxins detected in ASD children were always lower than those of their siblings, despite the comparable diet system and toxin exposure level, which the authors ascribed to a possible alteration in biotransformation and metabolism of the toxins.

Similarly, another cross-sectional study using a cohort of 233 children, comprising 172 autistic and 61 healthy (36 siblings and 25 non-parental) children, reported significant levels of mycotoxins in the urine and serum of the ASD children when compared to the controls, regardless of whether they were siblings or non-parental [24]. The evidence observed in this study revealed mycotoxins as a possible stress agent involved in the gene–environment interaction, eliciting ASD. In addition, the phenylalanine molecule of OTA may inhibit phenylalanine hydroxylase, an enzyme responsible for the catalysis of the hydroxylation of the aromatic side-chain of phenylalanine to produce tyrosine. In the absence of tyrosine, the biosynthesis of catecholamines is inhibited. Catecholamines interact with oxytocin, a hormone that plays a major role in social bonding and communication [23], thus contributing to the intellectual ability of an individual. This may explain the possible link of OTA exposure to autism, a syndrome associated with profound and irreversible intellectual disability.

However, whilst these studies showed evidence of an association between mycotoxins and ASD, an earlier study using a total of 54 children (25 ASD and 29 control) reported a contrary observation in spite of the wide range of urinary mycotoxins (87) examined [176]. The disparity in the studies as highlighted by De Santis et al. [23] may be due to small sample size (54 children) and poor sensitivity of the sample analysis (high limit of quantification). It is also noteworthy to mention that Duringer et al. [176] did not examine the genetic conditions of the patients, and thus may have included patients with other genetic syndromes. While there are still limited numbers of studies regarding to relationship between mycotoxin exposure and ASD, no single study was found in SSA on this issue, despite the high frequency and levels of mycotoxins in IYC foods in the region, as revealed in the present review. In SSA, research and policy that focus on disease prevalence, risk factors, and clinical intervention have been geared toward malaria, HIV, communicable diseases such as tuberculosis, and the reduction of infant mortality, a reason that may explain why there is limited information on neuro-developmental diseases in the region. A review by Franz et al. [177] highlighted the dearth of resource information on ASD in SSA, with the region having the least number of scientific articles, followed by North Africa and South America, notwithstanding the population of the region. It should also be noted that of the limited studies on ASD in SSA, no reliable studies on the prevalence of ASD were found [177]. Since Africa is predicted to contribute a good percentage to the world’s population of children, about 40%, by 2050 [178], it is likely that the reported figure of persons living with ASD worldwide is underestimated. This necessitates an urgent need for more studies in the region.

### 4.3. Causative Agent of Cancer

According to the International Agency for Research on Cancer (IARC), mycotoxins may exhibit carcinogenic properties on their host depending on the nature of the toxins. Amongst the most prevalent mycotoxins, AFs (AFB1, AFB2, AFG1, and AFG2) are classified as human carcinogens (Group 1) based on sufficient scientific evidence provided by epidemiological data and mechanistic studies. Group 2B includes OTA, AFM1, and FBs (FB1 and FB2) and is classified as possible carcinogens in humans based on scientific evidence for their carcinogenicity in animals, although data in humans are still non-conclusive [179].

Several epidemiological studies have reported an association between AFs and liver cancer, highlighting AFs as one of the major causative agents of hepatocellular carcinoma (HCC). AFs induces liver carcinogenesis through DNA adduct formation, leading to AGG to AGT mutations in codon 249 of the TP53 tumour suppressor gene (R249S) [180]. This mutation has been recorded in about 75% of HCC in regions of Asia and Africa with high incidence rates of AF contamination, in contrast to HCC cases reported in non-AF-contaminated regions [181,182]. A more difficult dimension of these toxins is their combination with the hepatitis virus, including hepatitis B virus (HBV) and hepatitis C virus (HCV), thus resulting in synergistic interaction and increasing the risk of individuals developing HCC [180,183]. A review by William et al. [165] and Kew [184] highlighted the high potency of AFs in hepatitis B surface antigen-positive subjects compared to persons without the virus. The risk of HCC increases from approximately 7% for patients with only HBV infection to approximately 60% for patients with HBV infection in combination with AF exposure [185]. Following the detailed risk assessment performed by the JECFA, a 0.3 cancer potency was estimated per year per 100,000 persons for an AFB1 exposure of 1 ng/kg bw/day in HBV-positive individuals, whereas a mean estimate of 0.017 was reported for HBV-negative individuals [186].

A study from Gambia probed the seasonal variation in R249S and HBV in relation to AFB1 exposure. Their findings showed a seasonal variation of the R249S mutation in circulating cell free DNA in the serum of subjects, thus reflecting seasonal variations in AF exposure and markers of HBV and the synergistic interaction between these risk factors. Within adult patients, other environmental factors, such as tobacco smoking and alcohol consumption, have also been reported to increase the risk of HCC [187]. HCC is rated as the third leading cause of cancer-related death globally owing to the inevitably poor prognosis resulting from metastasis and recurrence [188], but is ranked the sixth most common cancer worldwide [189], exerting an enormous burden in developing regions such as SSA [139]. In 2012, Liu et al. [190] systematically reviewed epidemiological studies on the associations between AF exposure and HBV infection as it relates to HCC in China and Taiwan, as well as SSA. Using a meta-analysis methodology to examine the population attributable risk (PAR) of AF-related HCC, their findings suggest that the PAR of AF-related HCC, which is the proportion of disease cases in the population that is avoidable if a risk factor is eliminated, was 17%, while for HBV-positive populations it was 21% and for HBV-negative populations it was 8.8% [190]. This signifies that the reduction in AF exposure in high-risk populations could potentially reduce HCC cases, thus preventing between 72,800 and 98,800 new cases of HCC. This hypothesis was supported by a Chinese study which investigated the impact of agricultural reforms substituting a highly AF-susceptible crop (maize) with rice as well as the implementation of universal HBV immunisation on the prevention of primary liver cancer [191]. The authors revealed that this approach yielded a remarkable estimated reduction rate of 65% in population mortality, with a decrease in the median concentration of AF-albumin of 19.3 pg/mg in 1989 to an undetectable level of <0.5 pg/mg in 2009.

Although childhood HCC is rare, it is the second most common malignant liver tumour after hepatoblastoma, accounting for less than 1% of childhood abdominal malignancies [192]. Earlier studies have reported quite a number of childhood HCC cases in SSA [193,194,195], mostly attributing the risk to HBV exposure. Admitting there is little or no information on the possible association between child AF exposure and risk of HCC in SSA, early exposure to carcinogens during the perinatal period, as revealed by the transplacental transfer of AFs, detection of AFM1 in breast milk, and frequency of these toxins in IYC food in the region are of concern. This is in line with a Kenyan study, which investigated the association between AF exposure and hepatomegaly in 218 children using an AF-albumin adduct biomarker and observed higher AF-albumin levels in hepatomegaly children as compared to normal children, suggesting a possible association between AF exposure and childhood hepatomegaly [196]. While there are limited studies on IYC, a Gambian study, using 494 adult individuals (97 cirrhosis individuals and 397 controls), examined the role of environmental and infectious exposure in cirrhosis [197]. The authors found a synergistic interaction between AFs and HBV, thus increasing the risk of cirrhosis. Interestingly, a significantly increased risk of cirrhosis was associated with lifetime consumption of groundnut, an AF-susceptible crop, as recorded by the authors [197].

In addition to AFs, OTA has been implicated to have a potential carcinogenic effect on humans. Though there is limited evidence of this effect in humans, OTA has demonstrated carcinogenicity, inducing renal tumours (adenomas and carcinomas), in several animal species (rats and mice), [198,199,200], and is thus classified as a Group 2B human carcinogen by the IARC. Marin-Kuan et al. [201] stated that the carcinogenicity of OTA may occur through a network of interacting epigenetic mechanisms, such as oxidative stress, protein synthesis inhibition, and the activation of specific cell signalling pathways. Studies have also shown that OTA alters gene expression, which resulted in affecting calcium homeostasis, thus disrupting the different pathways regulated by hepatocyte nuclear factor 4 alpha (HNF4α) and nuclear factor erythroid-2-related factor 2 (Nrf2) in the kidney [201,202]. Using a cross-sectional study design, Ibrahim et al. [203] investigated the relationship between OTA and HCC in 61 participants (39 HCC patients and 22 healthy controls) in Egypt. Although their findings revealed a strong association between OTA exposure and HCC, the limitations in the study design and small sample size limited the possible conclusion of OTA as a causative agent of HCC [202].

Furthermore, FBs have been associated with cancer of the oesophagus. Although several mechanisms have been postulated for their toxicity, FBs’ chemical structural resemblance to sphingosine and sphinganine has been hypothesised as the major molecular mechanism. The structure allows FBs to interfere with sphigolipid biosynthesis via N-acyltransferase, resulting in the accumulation of free sphingoid base, and the inhibition of ceramide synthase and an increase in sphingosine-1-phosphate that play opposing roles in mammalian cells [204,205]. FBs induce oxidative stress in oesophageal spindle-shaped N-cadherin (+) CD45 (-) osteoblastic (SNO) cancer cells through the generation of reactive oxygen species (ROS), thus causing damage to biological molecules [204]. In addition, FBs induce global DNA hypomethylation and histone demethylation, a possible alternative mechanism of FB carcinogenesis, as proposed by Chuturgoon et al. [206]. Thus far, there is still limited evidence, though studies from countries such as South Africa, Kenya, China, and Iran, with a high frequency of FB contamination and exposure, show a higher risk of oesophageal cancer [207].

Nevertheless, the possible co-occurrence and synergistic effects of mycotoxins should not be ignored. Fusaric acid, a neglected mycotoxin that often co-occurs with FBs, has also been reported as a cytotoxic agent to human oesophageal SNO cells. Fusaric acid induces oxidative stress and apoptosis in humans by significantly increasing Bax expression and caspase-8, -9, and -3/7 activities, while decreasing adenosine triphosphate (ATP) levels and Bcl-2 gene expression [208,209]. While cancer is rare in children, studies have linked exposures to environmental toxins in the perinatal period as a major causative agent of genetic diseases, such as cancer, in adulthood [210]. Thus, the frequency of mycotoxin occurrence in foods for infants and young children and mycotoxin exposure of this group, as highlighted in this review, should not be ignored [74,98,120,130,133,136].

## 5. Conclusion and Future Direction

The health and economical hazard posed by environmental contaminants such as mycotoxins cannot be overrated globally. However, due to frequent and multiple contaminations of staple diets in developing regions like SSA, there is a great concern about the adverse health impacts of mycotoxins on the population of this region. Unfortunately, IYC are not exempted in the surging adverse impact caused by mycotoxins and are the most vulnerable. A few available studies from SSA, as shown in the present review, highlight the occurrence and frequency of mycotoxins in IYC foods regardless of food type (complementary food, breast milk, or infant formula). This is may be linked to health effects shown in the exposure studies reported in the present review. Infant exposure to mycotoxins can occur as early asin utero and continue upon the introduction of complementary foods. While the reported occurrence and exposure level in the region seem to be high, which potentially lead to adverse health effects on IYC, such as impaired growth rate and immune and nervous system disorders, it is worrisome that only a few epidemiological studies have been conducted in SSA, mostly focusing on AFs and FBs. The scenario is exacerbated by the absence of functional regulatory, surveillance, and control measures to combat the apparent and hidden impacts of these toxins on child health.

Although numerous studies on mycotoxins originating from SSA have stated the need for a holistic approach toward protecting the population from the innate effects of mycotoxin exposure, the situation to date remains the same. There is a need to increase advocacy to engage researchers, farmers, food processors, clinicians, government, policy makers, and all stakeholders. Efforts should be made for detailed evaluations of the occurrence of mycotoxins in foods for infants and young children, as well as the exposure of this group. The utilisation of biomarkers in the exposure assessment should also be encouraged since it assesses the actual levels of exposure, taking into consideration all possible exposure routes. More epidemiological studies should be conducted to elucidate other possible health hazards that can occur as a result of exposure to mycotoxins while giving a better understanding of the relationship between mycotoxin exposure and health complications than already reported. Whilst more studies are required, it is important to emphasise the need to create awareness of mycotoxin occurrence, exposure, and possible health consequences in all communities. In addition, the need for SSA regional collaborative efforts similar to those in the European Union, the Association of Southeast Asian Nations, and Australia and New Zealand to develop policies and programmes to combat the menace caused by mycotoxin contamination is of paramount importance.

## Figures and Tables

**Table 1 foods-09-01585-t001:** Mycotoxin contamination of cereal- and oilseed-based complementary foods originating from sub-Saharan Africa.

Country	Toxin Type	Food Type	Sample Number	Sample Preparation	Analytical Method	LOD/LOQ (µg/kg)	% +ve	Range (µg/kg)	Reference
Kenya	^a^ ΣAF	Weaning flour	242	NA	TLC	NA	29	2–82	[37]
	^a^ ΣAF	Maize	186	NA	ELISA	NA	95	0–88.8	[44]
	^a^ ΣAF	Sorghum	89				100	0.1–194.4	
Nigeria	ΣFB	*Ogi*	30	SPE (C18), MulitiSep	LC-MS/MS	NA	93	3557 *	[9]
	DON						13	61 *	
	15-ADON						3	60 *	
	DON-3G						17	30 *	
	ZEN						3	39 *	
	α-ZEL						7	20 *	
	β-ZEL						10	19 *	
	HT-2						3	13 *	
	NIV						7	148 *	
	FUS-X						7	133 *	
	ΣFB	PSP	32	SPE (C18), MulitiSep	LC-MS/MS	NA	28	34–500	[48]
	DON					13.3	16	61–180	
	15-ADON					2.5	31	71–113	
	DON-3G					2.7	22	12–14	
	ZEN					3.6	81	27–388	
	α-ZEL					5.0	13	19–22	
	T-2					13.4	9	10–13	
	HT-2					4.4	25	22–35	
	FUS-X					32.9	22	66–276	
	NIV					48.1	22	70–104	
	DON	FMG	35	SPE (C18), MulitiSep	LC-MS/MS	7/14	11	<LOQ-55	[43]
	^a^ ΣAF					NA	26	<LOQ-17	
	ALT					40/80	9	<LOQ	
	HT-2					6.5/13	9	20–21	
	ΣFB					NA	77	42-3555	
Nigeria	STG	FMG	35	SPE (C18), MulitiSep	LC-MS/MS	1.3/2.5	29	4–7	[43]
	ZEN					3.3/6.5	9	<LOQ	
	ENN B					6.3/12	14	12–14	
	DON	FSG		SPE (C18), MulitiSep	LC-MS/MS	12/24	9	32–112	[43]
	^a^ ΣAF					NA	11	<LOQ–40	
	DAS					0.5/1.0	11	1–2	
	AME					6.3/12	23	30–35	
	ΣFB					NA	83	<LOQ–168	
	OTA					2.5/5	6	5–6	
	STG					2.5/5	23	<LOQ	
	3-NPA	Family cereal	26	Dilute and shoot	LC-MS/MS	0.8	61.5	1.6–18.5	[74]
	^a^ ΣAF					NA	100	0.4–11.1	
	ALT					0.4	15.4	0.4–0.9	
	BEA					0.008	69.2	0.1–0.6	
	CIT					0.16	88.5	1.2–151	
	DCIT					2	7.7	2.2–3.4	
	FA2					2.4	65.4	2.4–42.6	
	ΣFB					NA	100	62.3–1255	
	FB4					2.4	96.2	7.3–109	
	MON					1.6	92.3	1.7–34.8	
	OTA					0.4	7.7	0.5–0.5	
	3-NPA	Peanut	5	Dilute and shoot	LC-MS/MS	0.8	20	5.4	[74]
	^a^ ΣAF					NA	80	6.5–13.6	
	BEA					0.008	80	0.6–4.1	
	MON					1.6	60	2.2–3.5	
	3-NPA	*Ogi*	23	Dilute and shoot	LC-MS/MS	0.8	4.3	3.7	[74]
	^a^ ΣAF					NA	52.2	0.4–46.8	
	AFM1					0.4	4.3	0.9	
	BEA					0.008	82.6	0.1–5.3	
Nigeria	ALT	*Ogi*	23	Dilute and shoot	LC-MS/MS	0.4	4.3	0.4	[74]
	CIT					0.16	60.9	0.8-159	
	DCIT					2	4.3	3.9	
	FA1					2	43.5	1.2–11.3	
	FA2					2.4	43.5	5.3–42.3	
	ΣFB					NA	87.0	32.4–910	
	FB4					2.4	78.3	3.8-222	
	MON					1.6	17.4	2.4–32.3	
	OTA					0.4	8.7	0.7–1.8	
	ZEN					0.12	8.7	0.4–2.7	
	3-NPA	Tom bran	27	Dilute and shoot	LC-MS/MS	0.8	86.7	5.7–993	[74]
	Aflatoxicol					1	20	1.4–7.8	
	^a^ ΣAF					NA	83.3	0.5–590	
	AFM1					0.4	46.7	0.9–24.4	
	AFP1					2.4	3.3	14.6	
	ALT					0.4	30	1.2–7.2	
	BEA					0.008	96.7	0.1–69	
	CIT					0.16	73.3	1.7–1173	
	DON					1.2	6.7	30.8–31.6	
	DCIT					2	53.3	2.4–210	
	FA1					2	16.7	2.2–4.3	
	FA2					2.4	26.7	3.2–26.4	
	ΣFB					NA	96.7	7.8–1436	
	FB4					2.4	60	3.7–105	
	HFB1					1.6	10	2.1–8.1	
	MON					1.6	96.7	5.1–3450	
	NIV					1.2	10	11.4–23.8	
	OTA					0.4	26.7	0.5–26.4	
	OTB					1.6	10	3.5–113	
Nigeria	TeA	Tom bran	27	Dilute and shoot	LC-MS/MS	8	13.3	41.4–292	[74]
	ZEN					0.12	13.3	0.6–10.3	
	^b^ ΣAF	Cereal products	42	Dilute and shoot	LC-MS/MS	NA	21.4	1.0–16.2	[75]
	BEA					0.008/0.024	78.6	0.1–116	
	DON					1.2/3.6	4.8	52.9–61.5	
	ENN A					0.032/0.096	16.7	0.1–5.7	
	ENN B					0.024/0.072	19.0	0.03–6.9	
	ENN B1					0.04/0.12	14.3	0.4–14.3	
	ENN B2					0.04/0.12	7.1	0.09–0.34	
	FA1					2/6	2.4	2.0	
	FB1+FB2					NA	28.6	7.9–194	
	MON					1.6/4.8	7.1	7.2–10.6	
	ZEN					0.12/0.36	9.5	0.5–6.8	
Tanzania	ΣFB	Maize flour	191	NA	LC-FD	NA	69	21–3201	[38]
	^a^ ΣAF	Maize flour	41	NA	HPLC/FD	0.01-0.53	32	0.11–386	[72]
	DON			IAC	HPLC/UV	52	44	57–825	
	FB1+FB2			NA	HPLC/FD	47–53	83	63–2284	
	^a^ ΣAF	Maize flour	67	NA	HPLC/FD	0.01–0.53 (LOD)	58	0.33–69.5	[76]
	FB1+FB2					47–53 (LOD)	31	48–1225	
Zambia	AFB1	Peanut butter	96 ^c^	NA	ELISA	1	73	130 ^f^	[77]
	AFB1		252 ^d^	NA		1	80	10,740 ^f^	
	AFB1		606 ^e^	NA		1	53	1000 ^f^	
Zimbabwe	^b^ ΣAF	Peanut	18	IAC	HPLC/FD	NR	17	6.6–622.1	[78]
	^a^ ΣAF	Peanut butter	11		HPLC/FD	NR	90.9	6.1–241.2	

* Mean concentration of toxins in samples; ^a^ ΣAF = aflatoxin B1+B2+G1+G2; ^b^ ΣAF = AFB1+B2+G1; ^c^ 2012, ^d^ 2013, ^e^ 2014 survey study; ^f^ Maximum concentration; α-ZEL = α-zearalenol; β-ZEL = β-zearalenol; 3-NPA = 3-nitropropionic acid; 15-ADON = 15-acetyl-deoxynivalenol; AFM1, P1 = aflatoxin M1, P1; ALT = alternariol; AME = alternariol methyl ether; BEA = beauvericin; CIT = citrinin; DAS = diacetoxyscirpenol; DCIT = dihydrocitrinone; DON = deoxynivalenol; DON-3G = deoxynivalenol-3-glucoside; ELISA = enzyme-linked immunosorbent assay; ENN A, B, B1, B2 = enniatin A, B, B1, B2; ΣFB = fumonisin B1+B2+B3; FB4, A1, A2 = fumonisin B4, A1, A2; FMG = fermented maize gruel; FSG = fermented sorghum gruel; FUS-X = fusarenon-X; HFB1 = hydrolysed FB1; HPLC/FD = high-performance liquid chromatography/fluorescence detector; HT-2 = HT-2 toxin, IAC = immunoaffinity column; LC-FD = liquid chromatography/fluorescence detector; LC-MS/MS = liquid chromatography–tandem mass spectrometry; LOD = limit of detection; LOQ = limit of quantification; MON = moniliformin; NA = not available; NIV = nivalenol; OTA = ochratoxin A; OTB = ochratoxin B; PSP = processed soybean powder; SPE = solid phase extraction; STG = sterigmatocystin; TeA = tenuazonic acid; T-2 = T-2 toxin; TLC = thin layer chromatography; ZEN = zearalenone.

**Table 2 foods-09-01585-t002:** Mycotoxin contamination of infant formula, milk, and human breast milk originating from sub-Saharan Africa.

Country	Toxin Type	Food Type	Sample Number	Sample Preparation	Analytical Method	LOD/LOQ (µg/kg)	% +ve	Range (µg/kg)	Reference
Burkina Faso	AFB1	Infant formula	199	IAC	HPLC/FD	0.3/1	83.9	0–87.4	[73]
	FB1+B2					5/20	1.5	0–672.9	
	OTA					0.05/0.1	7.5	0–3.2	
Cameroon	AFM1	Breast milk	62	LLE	HPLC/FD	NA	5	0.005–0.625	[94]
Kenya	AFM1	Breast milk	165	NA	ELISA	0.005	74.6	<LOD–0.0475	[96]
	AFM1		39	NA	HPLC/FD	NA	15.4	0.0005–0.1527	
	AFM1	Milk	128	NA	ELISA	NA	100	0.002–2.56	[44]
Nigeria	AFM1	Breast milk	100	IAC	HPLC/FD	0.010	82	0.057 *	[92]
	BEA	Breast milk	75	QuEChERS	LC-MS/MS	0.006/0.011	56	<LOQ–0.019	[90]
	ENN B					0.004/0.009	9	<LOQ-0.009	
	OTA					0.048/0.096	15	<LOQ	
	AFM1					0.043/0.087	1	<LOQ	
	AFM1	Breast milk	120	LLE	HPLC/FD	NA	14	2–187	[45]
	AFM1	Breast milk	50	IAC	HPLC/FD	0.01/0.05	82	0.004–0.092	[91]
	AFM1	Breast milk	28	NA	TLC	2	18	LOD–4.0	[95]
	AFM1	Breast milk	40	NA	HPLC/FD	1	77	1–601	[97]
	BEA	Milk	36	Dilute and shoot	LC-MS/MS	0.008	27.8	0.04–0.4	[74]
	ZEN					0.12	2.8	0.6	
	3-NPA	Infant formula	17	Dilute and shoot	LC-MS/MS	0.8	11.8	19.6–22.5	[74]
	ΣAF					NA	5.9	4.6	
	ALT					0.4	5.9	0.7	
	CIT					0.16	5.9	3.6	
	BEA					0.008	17.6	0.1–13.4	
	DON					1.2	11.8	27.2–36	
	HT-2					3.2	5.9	18.8	
	MON					1.6	11.8	10–16	
	NIV					1.2	11.8	18.9–22	
	T-2					0.4	11.8	0.8–119	
Nigeria	ZEN	Infant formula	17	Dilute and shoot	LC-MS/MS	0.12	23.5	0.4–5.4	[74]
	AFM1	Breast milk	22	SPE	LC-MS/MS	0.002/0.004	18.2	0.002	[75]
	AME					0.0005/0.001	95.5	0.0005–0.0117	
	BEA					0.0001/0.0003	100	0.001–0.012	
	DCIT					0.014/0.028	27.3	0.014–0.0597	
	ENN B					0.0007/0.0014	72.7	0.0007–0.0101	
	ENN B1					0.0005/0.001	22.7	0.0005–0.0012	
	OTA					0.002/0.004	63.6	0.002–0.0676	
	OTB					0.0025/0.005	9.1	0.0053–0.0067	
	STG					0.0005/0.001	4.6	0.0012	
Sierra Leone	OTA	Breast milk	113	NA	HPLC/FD	0.2	35	0.2–337	[98]
	AFB1					0.05	18	0.05–372	
	AFM1					0.2	31	0.2–99	
	AFM2					0.07	62	0.07–77.5	
	AFG1					0.005	19	0.005–139	
	AFG2					0.003	22	0.003–366	
	Aflatoxicol					0.05	36	0.005–50.9	
Sudan	AFM1	Breast milk	94	LLE	HPLC/FD	0.013	54	0.007–2.561	[99]
Tanzania	AFM1	Breast milk	143	NA	HPLC/FD	0.005	100	0.01–0.55	[93]
	FB1	Breast milk	131	LLE, SAX	HPLC/FD	5.5/19.5	44	6.570–47.105	[100]

* Mean concentration of toxins in samples; AFB1, G1, G2, M1, M2 = aflatoxin B1, G1, G2, M1, M2; ΣAF = AFB1+B2;3-NPA = 3-nitropropionic acid; ALT = alternariol; AME = alternariol methyl ether; BEA = beauvericin; CIT = citrinin; DCIT = dihydrocitrinone; DON = deoxynivalenol; ELISA = enzyme-linked immunosorbent assay; ENN B, B1 = enniatin B, B1; FB1, B2 = fumonisin B1, B2; HPLC/FD = high-performance liquid chromatography/fluorescence detector; HT-2 = HT-2 toxin, IAC = immunoaffinity column; LC-MS/MS = liquid chromatography–tandem mass spectrometry; LLE = liquid–liquid extraction; LOD = limit of detection; LOQ = limit of quantification; MON = moniliformin; NA = not available; NIV = nivalenol; OTA = ochratoxin A; OTB = ochratoxin B; SAX = strong anion exchange; SPE = solid phase extraction; STG = sterigmatocystin; T-2 = T-2 toxin; TLC = thin layer chromatography; ZEN = zearalenone.

**Table 3 foods-09-01585-t003:** Summary of mycotoxin exposure studies on infants and young children in sub-Saharan Africa using occurrence and consumption data.

Country	Sample Size	Mycotoxin	Concentration (µg/kg)	Exposure Level, Mean (Range)	% Prevalence	PDI	TDI	Reference
						µg/kg bw/day		
**Nigeria**	70	AFB1 (infants)	324.7 *	1850 * ng/kg bw/day	NR	1.91 *	0.00017	[120]
		AFB1 (children)	324.7 *	740 * ng/kg bw/day		0.76 *	0.00017	
	137	AFB1	0.12–473.8	2.5–51,192 ng/kg bw/day	NR	NR	0.00017	[74]
		AFs	0.88–589.8	25.7–54,892 ng/kg bw/day			0.00017	
		FBs	4.6–1,540	0–138.6 ng/kg bw/day			2	
		OTA	0–26.4	0–2.03 ng/kg bw/day			0.1^a^	
		CIT	0.08–1,173	0.002–102 ng/kg bw/day			0.2	
		BEA	0.004–69	0–3.14 ng/kg bw/day			90	
		MON	0.8–3,450	0.02–156.8 ng/kg bw/day			200	
Tanzania	254	FBs (sorted maize)	19–1758	1.99 (0.32–144.29) ^b^ µg/kg bw/day	10 ^c^	<1.0	2.0	[121]
		FBs (unsorted maize)	19–21,666	36.80 (10.2–144.29) ^b^ µg/kg bw/day				
	143	AFs	0.33–69.47	0.14–120 ng/kg bw/day	39		0.017	[76]
		FBs	48–1224	0.005–0.88 µg/kg bw/day	21	<2	2	
	131	FBs	6.57–471.1	0.78–64.93 µg/kg bw/day	44.3		2.0	[93]
	143	AFM1	0.01–0.55	0:81–66.79 ng/kg bw/day	96 ^d^	NR	NR	[100]
	41	AFs	0.11–386	1–786 ng/kg bw/day	32		0.017	[72]
		DON	57–825	0.38–8.87 µg/kg bw/day	44		1	
		FBs	63–2284	0.19–26.37 µg/kg bw/day	83		2	
	191	FBs	21–3201	0.003–28.84 µg/kg bw/day	69	NR	2	[38]

* Mean concentration; ^a^ µg/kg bw per week; ^b^ FB exposure at 97th percentile; ^c^ exceeded the PMTDI; ^d^ exceeded 0.025 ng/mL(EU limit for AFM1 contamination in infant food (EC, 2006)); AFs = total aflatoxins; AFB1 = aflatoxin B1; BEA = beauvericin; CIT = citrinin; DON = deoxynivalenol; FBs = total fumonisins; MON = moniliformin; NR = not reported; OTA = ochratoxin A.

**Table 4 foods-09-01585-t004:** Summary of mycotoxin exposure studies on infants and young children in sub-Saharan Africa using biomarkers (blood (pg/mg) and urine (ng/mL)).

Country	Sample Size	Matrix	Mycotoxin	Concentration: Mean (Range)	% Prevalence	PDI	TDI	Reference
						µg/kg bw/day		
Benin	200	Blood (sampled in February)	AFs	37.4	98-100	NR	NR	[34]
		Blood (sampled in June)	AFs	38.7				
		Blood (sampled in October)	AFs	86.8				
Benin and Togo	480	Blood	AFs	32.8 (5–1064)	99	NR	NR	[33]
		Blood (fully weaned)	AFs	45.6 (38.8–53.7)				
		Blood (partially breastfed)	AFs	18.0 (15.2–21.3)				
Benin and Togo	479	Blood	AFs	32.8 (25.3–42.5)	99			[126]
Gambia	119	Blood (maternal)	AFs	40.4 (4.8–260.8)	100	NR	NR	[36]
	99	Blood (cord)	AFs	10.1 (5.0–189.6)	48.5			
	118	Blood (infant)	AFs	8.7 (5.0–30.2)	11			
Guinea	305	Blood (at harvest)	AFs	12.7 (10.9–14.7)	88.2	NR	NR	[127]
	288	Blood (post-harvest)	AFs	16.3 (14.4–18.5)	93.4			
Tanzania	166	Blood (at recruitment)	AFs	4.7 (3.9–5.6)	67	NR	NR	[128]
		Blood (after 6 months)	AFs	12.9 (9.9–16.7)	84			
		Blood (after 12 months)	AFs	23.5 (19.9–27.7)	99			
Uganda	96	Blood	AFs	9.7 (8.2–11.5)	95.8	NR	NR	[129]
Cameroon	220	Urine	FB1	2.96 (0.06–48)	11	0.71	2.0	[130]
			DON	2.22 (0.1–77)	17	2.89	1.0	
			OTA	0.2 (0.04–2.4)	32	NR	NR	
			AFM1	0.33 (0.06–4.7)	31			
			ZEN	0.97 (0.65–5.0)	4			
			α-ZEL	0.98 (0.26–1.3)	4			
			β-ZEL	1.52 (0.02–12.5)	8			
Ethiopia	200	Urine	AFM1	0.064 (0.06–0.07)	7	NR	NR	[131]
			AFB2	0.047 (<LOQ–0.063)	4.5			
			AFG1	0.061 (0.054–0.065)	2.5			
			AFG2	0.0068 (0.066–0.07)	3			
Guinea	50	Urine	AFB1	0.027 (0.016–0.043)	16	NR	NR	[132]
			AFB2	0.0008 (0.0005–0.0013)	58			
			AFG1	0.027 (0.023–0.031)	2			
			AFG2	0.0011 (0.0007–0.0017)	36			
			AFM1	0.0163 (0.0101–0.027)	64			
Nigeria	19	Urine	AFM1	0.1	10.53	0.67	NA	[133]
			DON-15-O-GLU	1.5	5.3	NR	NA	
			FB1	3.7	21.1	NR	2.0	
			OTA	0.1	21.1	NR	0.12 ^a^	
Tanzania	166	Urine(at recruitment)	FBs	0.314 (0.257–0.383)	98	NR	NR	[128]
		Urine (after 6 months)	FBs	0.167 (0.135–0.207)	96			
		Urine (after 12 months)	FBs	0.570 (0.465–0.698)	100			
	166	Urine (at recruitment)	DON	1.1 (0.8–1.4)	51	0.063	1	[134]
		Urine (after 6 months)	DON	2.3 (1.7–3.2)	70	0.122	1	
		Urine (after 12 months)	DON	5.7 (4.1–7.9)	80	0.268	1	

^a^ µg/kg bw per week; AFs = total aflatoxins; AFB1, B2, G1 G2, M1 = aflatoxin B1, B2, G1, G2, M1; DON = deoxynivalenol; DON-15-O-GLU = deoxynivalenol-15-O-glucuronide; FB1 = fumonisin B1; LOQ=limit of detection; NA = not applicable; NR = not reported; OTA=ochratoxin A; ZEN=zearalenone; α-ZEL=α-zearalenol; β-ZEL= β-zearalenol.

**Table 5 foods-09-01585-t005:** Summary of studies on relationships between mycotoxin exposure and child growth in sub-Saharan Africa.

Country	Population Size/Group	Matrix/Study Design	Toxin Type	Concentration Range (Rate)	% Rate of Malnutrition	Result	Reference
Benin and Togo	480 Children Age: 9 months–5 years	AF-albumin (blood)/cross-sectional	AFs	5–1064 pg/mg (99%)	HAZ: 33% WAZ: 29% WHZ: 6%	AF-albumin concentration negatively correlated with the growth parameters (HAZ (*p* = 0.001), WAZ (*p* = 0.005), and WHZ (*p* = 0.047)).	[33]
Benin	200 Children Age: 16–37 months	AF-albumin (blood)/longitudinal (8 months) at three points: February, June, and October	AFs	February: 37.4 pg/mg (98%) June: 38.7 pg/mg (99.5%) October: 86.8 pg/mg (100%)	Not calculated	AF-albumin level measured at recruitment or the mean of three time points was inversely associated with HAZ and WHZ scores measured at the last time point. Strong negative correlation (*p* < 0.0001) between AF-albumin and height increase over the 8 months follow-up after adjustment for age, sex, height at recruitment, socioeconomic status, village, and weaning status. Highest quartile of AF-albumin was associated with a mean 1.7 cm reduction in growth over 8 months compared with the lowest quartile.	[34]
Cameroon	220 Children Age: 1.5–4.5 years	Urine/cross-sectional	OTA DON AFM1 FB1 ZEN β-ZEL α-ZEL	73% of the samples were contaminated with mycotoxin	HAZ: 39% WAZ: 37% WHZ: 23%	Significant differences were observed between the weaning categories and AFM1 concentration detected in the urine. No association between the different malnutrition categories (stunted, wasting, and underweight) and the mycotoxin concentrations detected in the urine samples.	[130]
Ethiopia	200 Children Age: 1–4 years	Urine/cross-sectional	AFs	Values in ng/mL: AFM1: 0.06–0.07 AFB2: <LOQ–0.06 AFG1: 0.05–0.065 AFG2: 0.066-0.07 AF (17%)	HAZ: 45% WAZ: 17% WHZ: 1%	No association between the different malnutrition categories (stunted, wasting, and underweight) and AF exposure.	[131]
Gambia	138 Children Age: 0–12 months	AF-albumin (blood)/longitudinal (14 months follow-up from birth until one year of age)	AFs	Maternal blood: 4.8–260.8 pg/mg (100%) Cord blood: 5.0–189.6 pg/mg (48.5%)	Not calculated	High maternal AF-albumin was associated to lower HAZ (*p* = 0.044) and WAZ (*p* = 0.012) scores. Reduction of maternal AF-albumin from 110 pg/mg to 10 pg/mg would lead to an increase of 0.8 kg in weight and 2 cm increase in height of a child within the first year of life.	[36]
				Infant blood: 5.0–30.2 pg/mg (11%)		AF-albumin measured at Week 16 was negatively related to HAZ (*p* = 0.002).	
	374 Infants Age: 0–2 years	AF-albumin (blood)/longitudinal	AFs	48, 98, and 99% had detectable AF-albumin concentrations (LOD > 3.0 pg/mg at 6, 12, 18 months, respectively)	HAZ: 25.9% WAZ: 24.4% WHZ: 12.9%	Inverse relationships between AF-albumin adducts and HAZ, WAZ, and WHZ scores from the age of 6 to 18 months. Inverse relationship between AF-albumin at 6 months and change in WHZ between 6 and 12 months (*p* = 0·013). AF-albumin at 12 months was associated with changes in HAZ and infant length between the age of 12 and 18 months. AF-albumin at 6 months was associated with IGFBP-3 at 12 months (*p* = 0·043).	[141]
Kenya	242 Children Age: 3–36 months	Weaning flour/cross-sectional	AFs	2–82 µg/kg (29%)	HAZ: 34% WAZ: 30% WHZ: 6%	Highly significant association between children that consumed AF-contaminated flour and prevalence of wasting (*p* = 0.002).	[37]
	204 Children Age: 1–3 years	Cereals (maize, sorghum) and milk/cross-sectional	AFs AFM1	AF: 0–194 µg/kg in cereals AFM1: 0.002–2.56 µg/kg in milk (98%)	HAZ: 41% WAZ: 17% WHZ: 4%	AFM1 was negatively associated with HAZ (*p* = 0.047). No association between total AFs (AFB and AFG) and HAZ, WAZ, and WHZ Scores.	[44]
Kenya	881 Children Age: 0–2 years	AFB1-lysine adduct/cluster randomised longitudinal	AFs	*18.1 pg/mg albumin	Not calculated	The intervention significantly reduced endline serum AFB1-lysine adduct levels (*p* = 0.025). No effect on the prevalence of stunting or LAZ, though a significant effect on child linear growth was found at midline (11–19 months).	[142]
Nigeria	58 Children Age: 6–48 months (with severe acute malnutrition)	AFB1-lysine/cross-sectional	AFs	0.2–59.2 pg/mg albumin	Severe acute malnutrition: 81% HAZ: 74%	Significantly higher AFB_1_-lysine concentrations in stunted children compared to non-stunted children as well as in children with severe acute malnutrition compared to controls. No significant association between AFB1-lysine and stunting after adjustment for malnutrition status (OR = quartile 3, 1.21; 95% CI: 0.086–31.45) and no correlation between AFB1-lysine and WAZ.	[143]
Tanzania	215 Infants Age: 6 months	Maize/longitudinal (follow-up at 6 and 12 months of age)	FBs	21–3201 µg/kg (69% of 191 maize samples)	Not calculated	Infants exposed to FBs above the PMTDI (2 µg/kg) were significantly shorter by 1.3 cm and lighter by 328 g at 12 months (*p* = 0.002).	[38]
	143 Infants Age: 0–6 months 143 Lactating mothers	Breast milk/longitudinal (follow-up at three points: 1st, 3rd, and 5th months of age)	AFM1	1st month: 0.01–0.55 ng/mL 3rd month: 0.01–0.47 ng/mL 5th month: 0.01–0.34 ng/mL	1st month: HAZ: 11% WAZ: 4% WHZ: 4% 3rd month: HAZ: 13% WAZ: 9% WHZ: 1% 5th month HAZ: 17% WAZ: 10% WHZ: 3%	Significant inverse association between AFM1 exposure levels and WAZ or HAZ (*p* < 0.05).	[100]
Tanzania	166 Children Age: 6–14 months old	AF-albumin and UFB1/ longitudinal (12 month follow-up)	AFs FBs	AF-albumin: at recruitment: *4.7 pg/mg (67%), 6 months: *12.9 pg/mg (84%), 12 months: *23.5 pg/mg (99%) UFB1: at recruitment: *313.9 pg/mg (98%), 6 months: *167.3 pg/mg (96%), 12 months: *569.5 pg/mg (100%)	At recruitment HAZ: 44% WAZ: 8% WHZ: 2% At 6 months: HAZ: 55% WAZ: 14% WHZ: 2% At 12 months: HAZ: 56% WAZ: 14% WHZ: 0.7%	No significant negative association between mean AF-albumin levels and child growth (HAZ, WHZ, or WAZ score). Negative association between mean UFB1 concentrations (at recruitment, and 6 and 12 months from recruitment) and HAZ at recruitment.	[128]
	143 Infants Age: under 6 months	Maize flour/longitudinal (follow-up at three points: 1st, 3rd, and 5th months of age)	AFs FBs	AF: 0.33–69.5 µg/kg (58% of 67 maize samples) FB: 48–1224 µg/kg (31% of 67 maize samples)	WAZ: 35% of 115 infants HAZ: 43% of 115 infants	Insignificant association was observed between exposure to AFs or FBs and stunting or underweight.	[76]
	300 Children	Maize/cluster randomised controlled trial	AFs FBs	Not reported	Not calculated	AF and FB intakes were inversely associated with WAZ. WAZ was 6.7% lower in the intervention group. Mean WAZ difference between the groups was 0.57 (*p* = 0.007).	[144]
Tanzania	114 Children Age: under 36 months	AFB1- lysine and UFB1/ longitudinal	AFs FBs	AFB1-lysine: 0.28–25.1 pg/mg (72%) UFB1:<LOD–16.6 ng/mL (80%)	At 24 months: HAZ: 61% WAZ: 17% WHZ: 3% At 36 months: HAZ: 75% WAZ: 21% WHZ: 0%	No associations were found between AF exposure and growth impairment as measured by stunting, underweight, or wasting. However, FB exposure was negatively associated with underweight.	[41]
Uganda	246 Dyads	Maternal serum/longitudinal (follow-up: pregnancy up to one year)	AFs	AFB1: *113.9 pg/mg	Not calculated	A negative effect of AF exposure on infant linear growth (HAZ) in HIV-positive pregnant women and their infants. Infants of HIV-positive women were in high perinatal AF category with lower HAZ scores (0.460) when compared with the infants of HIV-negative low-AF exposed women (*p* = 0.006).	[145]
	220 infants Age: 0–48 h	Maternal serum/cross-sectional	AFs	AFB1: 0.71–95.6 pg/mg albumin	Not calculated	Maternal AFB-lysine levels were significantly associated with lower birth weight (adj-β: *p* = 0.040), lower WAZ (adj-β: *p* = 0.037), smaller HC (adj-β: *p* = 0.035), and lower HCZ (adj-β: *p* = 0.023) at birth. No significant associations were observed between maternal AFB-lysine levels and infant length, WHZ, HAZ, or gestational age at birth.	[146]

* Mean concentrations, AFs = total aflatoxins; AFB1, M1 = aflatoxin B1, M1; FBs = total fumonisin B; FB1 = fumonisin B1; IGFBP-3 = insulin-like growth factor-1 binding protein-3; DON = deoxynivalenol; ZEN = zearalenone; α-ZEL = α-zearalenol; β-ZEL = β-zearalenol; LOD = limit of detection; LOQ = limit of quantification; OTA = ochratoxin A; UFB1 = urinary fumonisin B1; ZEN = zearalenone; adj-β = adjusted model; HAZ = height-for-age Z-score; HC = head circumference; HCZ = head circumference-for-age Z-score; WAZ = weight-for-age Z-score; WHZ = weight-for-height Z-score.

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
