# Peer review of "Mycotoxin Occurrence, Exposure and Health Implications in Infants and Young Children in Sub-Saharan Africa: A Review"

_foods, 2020, doi:10.3390/foods9111585_

Round 1

Reviewer 1 Report

The review is focused on the evaluation of the presence of different types of mycotoxins in sub-saharian Africa. It covers the most relevant toxins as well as their implications/ relationship with different types of diseases and alterations of human health. I believe the review will result of interest for people working in the field, and can provide an overview for the general public of the situation in this particular geographic area.

[*] What is the main question addressed by the research? Presence/ situation of mycotoxins in SSA.

[*] Is it relevant and interesting? Yes, the authors provide an overview of mycotoxins in SSA and their impact on children who among the most susceptible.

[*] How original is the topic? NA, it is a review.

[*] What does it add to the subject area compared with other published material? The authors gathered many studies dealing with the presence of mycotoxins in the region of interest, covering many different areas, from surveillance in foods and population, to health implications (relationship with different types of diseases and health problems).

[*] Is the paper well written? Yes

[*] Is the text clear and easy to read? Yes

[*] Are the conclusions consistent with the evidence and arguments presented? Yes

[*] Do they address the main question posed? Yes

Author Response

Dear Reviewer,

Thank you for the positive comments.

Reviewer 2 Report

The manuscript is interesting and well-written. I read the article with pleasure and interest

However, the Authors should consider the following suggestions from the reviewer:

  • the table with the most common mycotoxins , their chemical formula, occurrence and main impact on the living organisms in the first part of the manuscript seems to be necessary
  • Mycotoxins are also present in food of animal-origin (also in sub-Saharan Africa) for example in fish, which are also consumed by children. Information about this fact should be added
  • the list of abbreviations used in the text would increase the readability of the manuscript.

Author Response

Dear Reviewer,

Thank you for the insightful comments.

Reviewers comment

However, the Authors should consider the following suggestions from the reviewer:

the table with the most common mycotoxins , their chemical formula, occurrence and main impact on the living organisms in the first part of the manuscript seems to be necessary

Answer

We did not include tables containing information as highlighted because several review articles have reported this information.

Reviewers comment

Mycotoxins are also present in food of animal-origin (also in sub-Saharan Africa) for example in fish, which are also consumed by children. Information about this fact should be added

Answer

Animal products are not exempted from mycotoxin contamination. However, we did not report this in our review because there are no available data on the occurrence of mycotoxins in animal-based infant foods in sub-Saharan Africa.

Reviewers comment

The list of abbreviations used in the text would increase the readability of the manuscript.

Answer

We did not include list of abbreviation because it is not stated in the journal guideline (“Abbreviations should be defined in parentheses the first time they appear in the abstract, main text, and in figure or table captions and used consistently thereafter”).

Reviewer 3 Report

This review puts the focus on the occurrence of mycotoxins in the SSA region in view of a possible holistic approach of protection of the population (especially children). The language used is very clear and the collected results are of interest although parts (of course) have been published in other reviews. Very detailed results are given in the overview. It would be of interest to add also some refs. on the methods used to analyze these mycotoxins [Rapid Commun. Mass Spectrom., 20, 771-776 (2006)] in order to give the reader a more complete state of the art and to allow the reader to embark in further research on this theme in various directions. In the introductory part, the state of the art on mycotoxins would become more complete if the problem of the sick building syndrome is touched because not only food and environment do play a role in the issue of mycotoxins [Building and Environment, 46, 945-954 (2011)]. Adding a few sentences and appropriate references would be sufficient. Concerning the biomarker analysis (part 3.2) it is recommended to add some references on this topic of identification of volatiles produced by fungi [Fungal Biology, 116, 941-953 (2012); Sci. Total Environment, 414, 277-286 (2012)].

Author Response

Dear Reviewer,

Thank you for the insightful comments.

Reviewers comment

It would be of interest to add also some refs. on the methods used to analyze these mycotoxins [Rapid Commun. Mass Spectrom., 20, 771-776 (2006)] in order to give the reader a more complete state of the art and to allow the reader to embark in further research on this theme in various directions.

Answer

The analytical methods used to analyse mycotoxins as well as their references are listed in Table 1.

Reviewers comment

In the introductory part, the state of the art on mycotoxins would become more complete if the problem of the sick building syndrome is touched because not only food and environment do play a role in the issue of mycotoxins [Building and Environment, 46, 945-954 (2011)]. Adding a few sentences and appropriate references would be sufficient.

 Answer

The environment we referred in our manuscript comprise of both indoor and outdoor. We did not expatiate exposure by environment because there are little or no information on this aspect in sub-Saharan Africa.

Reviewers comment

Concerning the biomarker analysis (part 3.2) it is recommended to add some references on this topic of identification of volatiles produced by fungi [Fungal Biology, 116, 941-953 (2012); Sci. Total Environment, 414, 277-286 (2012)].

Answer

Our review is focused on the available data in sub-Saharan Africa. The studies recommended are outside the geographical scope of our manuscript.